# Effectiveness of a Health Behavioural Intervention Aimed at Reduction of Risky Sexual Behaviours among Young Men in the KwaZulu-Natal Province, South Africa

**DOI:** 10.3390/ijerph16111938

**Published:** 2019-05-31

**Authors:** Thabang Manyaapelo, Bart Van den Borne, Robert A. C. Ruiter, Sibusiso Sifunda, Priscilla Reddy

**Affiliations:** 1Social Aspects of Public Health (SAPH) Research Programme, Human Sciences Research Council, Private Bag X41, Pretoria 0001, South Africa; ssifunda@hsrc.ac.za (S.S.); preddy@hsrc.ac.za (P.R.); 2Department of Health Education & Health Promotion, Maastricht University, P.O. Box 616, 6200 Maastricht, The Netherlands; b.vdborne@maastrichtuniversity.nl; 3Department of Work & Social Psychology, Maastricht University, P.O. Box 616, 6200 Maastricht, The Netherlands; r.ruiter@maastrichtuniversity.nl; 4Child and Family Studies, Faculty of Community and Health Science, University of the Western Cape, Cape Town 7535, South Africa

**Keywords:** risky sexual behaviour, HIV/AIDS, behavioural intervention, theory of planned behaviour, substance use, condom use

## Abstract

Two studies evaluating the same behavioural intervention were conducted in two areas in the KwaZulu-Natal province of South Africa using a randomized pre-test post-test control group design for study 1 (peri-urban) and a pre-test post-test design without a control group for study 2 (rural). The intervention included discussions and skills training on: (1) notions of masculinity, manhood, and responsibility, (2) personal and sexual relationships, (3) general communication skills, and (4) alcohol and other substance use. The intervention was aimed at men between 18 and 35 years of age. Measures of attitude, subjective norms, perceived behavioural control and intention for condom use, human immunodeficiency virus (HIV) testing, reduction of alcohol and drug use, avoiding sex while intoxicated, and avoiding sex with intoxicated people were assessed using a facilitator-administered questionnaire. The results for study 1 showed that 4 of the 19 variables scored significantly different at baseline and that all 19 variables showed no significant changes between pre-test and post-test. For study 2, one significant difference was found for attitude towards avoiding sex when one is intoxicated. Overall, the intervention had minimal success with just one area of positive effect. Further development and testing of this programme is recommended before it can be considered for broader scale implementation.

## 1. Introduction

South Africa has the largest population of people infected with HIV, at 7.9 million people [1], and the province of KwaZulu-Natal has the highest rate of human immunodeficiency virus (HIV) infection in the country [2]. Risky sexual behaviours remain one of the major contributors to new HIV infections and continue to hamper prevention strategies. Key risky behaviours include multiple concurrent sexual partnerships, non or incorrect and inconsistent use of condoms, and engaging in sexual activities while intoxicated or under the influence of illicit substances [3,4,5,6]. It is important that these behaviours are not observed in isolation from each other. For instance, people who report having more than one sexual partner are more likely to use condoms compared to those in monogamous relationships, they tend to have their sex debut at a younger age, have a greater number of lifetime sex partners, and are less likely to have been tested for HIV [5,7,8]. There is also evidence showing that inconsistent condom use and multiple sexual partnerships can be attributed to the use of alcohol and other drugs [9,10]. This is particularly important for sexually transmitted infections (STI) and HIV transmission since alcohol consumption can affect sexual decision making through its effect on higher executive control functions such as time estimation, attention, planning, decision-making, and inhibitory control [11]. 

In a recent study on condom use among young people in South Africa, a surprising finding was that the perceived risk of HIV infection was found to not be a significant predictor of consistent condom usage [12]. Condom use for the 15–24 year old age groups was about 49% for women and 67% for men in both the 2012 and 2017 HIV national surveys, suggesting that condom usage had remained largely unchanged over this time period [1]. There was however a condom use increase for the 25–49 year old group over the same period, with a 3.3% increase for women and a 4.1% increase for men [1]. These findings on recent trends in condom use behaviour among the South African population suggest that more efforts are needed to increase condom use among young people to enhance HIV prevention. Globally, the decline in new infections has remained static since 2010, with some of the countries reaching the set targets while other countries struggle. It is evident that all known strategies in HIV prevention should be employed if the goal to end acquired immunodeficiency syndrome (AIDS) by 2030 is to be achieved [13,14]. The present study is such an attempt to assist in HIV prevention efforts, specifically targeting young men. 

In Sub-Saharan Africa several STI/HIV-focused interventions covering different aspects of HIV prevention have been tested, which include but are not limited to: school-based interventions [15,16], interventions to increase antiretroviral therapy (ART) adherence [17], interventions to increase male participation in prevention of mother-to-child transmission (PMTCT) [18] and to reduce HIV stigma in PMTCT [19], interventions that reduce the use of alcohol and other drugs [20,21,22], and interventions that link HIV-infected people to care, therefore opening room to initiate ART [23]. These latter interventions showed that home-based HIV testing and counselling, together with a proper referral plan, aid considerably in reducing the time it takes to successfully start people on care protocols. When looking at school-based interventions, it is clear that these programmes are best placed to reach young people, who are currently among the highest-infected groups. The evidence however further suggests that school-based interventions only have a positive effect on condom use and not on STIs [16]. Interventions which sought to reduce alcohol and drug use reported decreased incidents of engaging in sex while intoxicated and a reduction in the number of sexual partners [20,22]. 

The current study used the theory of planned behaviour (TPB) as the guiding framework in determining the target points for a health behaviour intervention aimed at reducing risky sexual behaviours in young men from a highly vulnerable population. The theory of planned behaviour has been used to investigate a wide range of behaviours such as protective behaviours (supplement use, blood donation, sun protection), risk behaviours (alcohol use, illicit drug use, smoking cessation), and detection behaviours (cervical and breast cancer screening) [24]. The theory of planned behaviour (for a recent re-formulation, see Fishbein and Ajzen [25]) posits that behavioural intention is determined by three evaluative constructs: attitude, subjective norm, and perceived behavioural control. Attitude represents a person’s overall evaluation of the anticipated outcomes (favourable versus unfavourable) of the behaviour. Subjective norm is the perceived social pressure to engage or not engage in a behaviour. Perceived behavioural control is people’s perceptions of their ability to perform a given behaviour [26]. 

The TPB has been able to explain varying proportions of the variance with different cognitive measures contributing differently for the given studies. In the most comprehensive meta-analysis of TPB interventions to date, covering 82 articles reporting on 123 interventions [27], a few key things were noted: the effect sizes ranged between 0.14 and 0.68 in the cognitive constructs (attitude, subjective norm, perceived behavioural control, intention, behavioural, normative and control beliefs), for changes in behaviour the mean effect size was 0.5. Furthermore, interventions conducted in public with groups were more effective than those conducted in private for individuals, and gender, education, and behavioural domain are moderators for intervention effectiveness. 

The aim of this study was to evaluate the effectiveness of a behavioural intervention to reduce risky sexual behaviours of non-condom use, not testing for HIV, use of alcohol and drugs, engaging in sexual activities while intoxicated or with a partner who is intoxicated with alcohol or drugs, in a population of young men between the ages of 18 and 35 in the KwaZulu-Natal province. It was hypothesized that the intervention would lead to an increase in attitudes, subjective norms, and perceived behavioural control towards reducing those risky sexual practices, which in turn would lead to an increase in intentions towards reducing the risky sexual behaviours. Intention has been shown to be the proximal antecedent to behaviour [28]. The behaviours themselves were not measured directly.

## 2. Materials and Methods

### 2.1. Study Areas

The study was conducted in both a peri-urban (study 1) and a rural area (study 2). The peri-urban area of Clermont is located in the KwaZulu Natal province, roughly 30 km from the eastern coast city of Durban. Clermont is a densely populated area, including approximately 31,600 households, with a majority Black African population. In South Africa, the official population group classifications are Black African, White, Asian/Indian, and Coloured. Black African is the population classification designated for the indigenous groups in South Africa. Only 32% of the residents were employed. Housing infrastructure is characterized by a mixture of free-standing dwellings, shacks, and hostels covering a small area of about 13 km^2^ [29]. The shacks and hostels are indicative of the migration that is common in South Africa between rural and peri-urban areas. People usually come to the peri-urban centres like Clermont in search of employment opportunities, and usually end up living in informal settlements and migrate a few times a year between urban and rural homesteads [30]. 

The rural area of Nkungumathe in the KwaZulu-Natal province also has a majority Black African population. Nkungumathe is located approximately 250 km from Durban. The area had a minimal road infrastructure in 2010/2011 when the data were collected. The infrastructure in this area was characterized by sparsely located rural dwellings spanning a vast hilly topography. The majority of the housing structures were a combination of mud, thatch, brick, and stick and only a few houses were constructed of brick and tiles. Less than 10% of the residents were employed at the time of the survey.

### 2.2. Research Design

For study 1 in the peri-urban area, a randomized pre-test post-test control group design was applied. The eligibility criteria were age (18–35), they had to reside in the area under study, they had to be isiZulu speaking, and they had to be available for the follow-up study in 3 to 4 months after the pre-test. The participants were expected to attend a total of 4 or 6 visits to complete the study. At visit 1, all of the recruited participants who met the inclusion criteria were randomized into either control or experiment groups after completing the pre-test baseline questionnaire. At visit 2, all of the participants would start their first session. At visit 3, all of the participants attended their second session. At visit 4, the control group completed the post-test questionnaire while the experiment group attended their third session. At visit 5, the experiment group attended the fourth session. At visit 6, the experiment group completed the post-test questionnaire. A total of 428 participants were randomly assigned to either control or experimental groups. A total of 192 participants was assigned to the control group while 236 were assigned to the experimental group. In the follow-up measurement, 129 participants in the control and 182 participants in the experiment group took part. 

For study 2 in the rural area, a pre-test post-test design without a control group was applied due to structural limitations of available research funding and participant accessibility. Therefore, an area of limited size of approximately 9 km^2^ with a limited number of 178 households was selected for the study. This area was selected because the research team previously established contact with the community with the intention of a community development project. The participants were expected to attend a total of 6 visits to complete the study. At visit 1, the recruited participants who met the inclusion criteria completed the pre-test baseline questionnaire. At visit 2, the participants started with the first session. At visit 3, the participants attended their second session. At visit 4, the participants attended the third session. At visit 5, the participants attended the fourth session. At visit 6, participants completed the post-test questionnaire. A total of 147 participants met the eligibility criteria and took part in the baseline questionnaire. Only 128 participants were able to return for the follow up questionnaire. A preliminary participant censor-type survey had been conducted a few months before. This survey estimated that there were about 400 young men who could be eligible to participate in the study. Due to work and schooling commitments outside the area, the 400 eligible participants estimate was further reduced in terms of the total who could actually participate. This relatively low number of participants led to the decision to include all of the eligible participants in an experimental condition.

### 2.3. Intervention Curriculum and Control Curriculum

The curriculum was based on an intervention programme developed for soon-to-be-released prison inmates in South Africa [31,32] but was adapted for the communities targeted for study. The process of adaptation involved preliminary qualitative interviews in the form of focus group discussions in each of the two study areas. These interviews were among community leaders in the respective areas, mothers and fathers, as well as young men. A community survey was also conducted in the rural area to gain a better understanding of the local context. The final product was an adapted intervention curriculum called Ubudoda Abukhulelwa Responsible Manhood: towards the Development of a Culturally Tailored and Contextually Sensitive Life Skills Programme for Heterosexual Men in South Africa.

Objectives of the intervention were firstly, to reduce risky sexual behaviours such as lack of condom use, non-testing for HIV, alcohol and drug use. Secondly, to encourage participants to avoid sex when personally intoxicated, and avoid sex with intoxicated people, and lastly, to encourage positive male supportive roles within relationships. The intervention included knowledge building and understanding, skills training, confidence building, and communication training and applied strategies of role playing, critical dialogue and reflection, and peer support. The intervention was delivered by peer educators to enhance learning and skills training. The intervention comprised of 4 sessions which were 3 hours long each and was administered over a 4-week period on the following themes:Notions of masculinity, manhood, and responsibilityPersonal and sexual relationshipsGeneral communication skillsAlcohol and other substance use

In each session, two hours were used for knowledge transfer, discussion, and dialogue, and one hour for skills training (see Appendix B). The aim of each session was to introduce the topic and have an in-depth discussion around the above-mentioned themes, using the probes in the learning objectives as a guide. The discussion was facilitated by two peer educators per session, taking turns to guide the discussion. The participants (8–12 per session) sat in a circle with the peer educators sitting in between them. The conclusion for each session was to teach a skill that would influence a behaviour change for the targeted area. For example, for the topic about condom use, the skills training would include a practical demonstration by the peer educators, using models, on how to correctly wear a condom. The participants would be given an opportunity to practice this skill on the models until they gained enough confidence to do it on their own without instruction. The confidence building communication sessions included a role play session where participants were given imaginary scenarios and asked to demonstrate how they would overcome them. Each participant would be given an opportunity to role play until they felt confident to initiate and execute on their own.

The men in the comparison condition (control) in the peri-urban study watched a 96-minute-long South African-made film called Yesterday (2004), directed by Darrell Roodt and starring Ms. Leleti Khumalo. The film, located in a rural village, chronicles the life of a Zulu mother who contracts HIV from her migrant miner husband. The story continues with how she copes with the diagnosis while raising her young daughter. The second session for the control group was an audio recording by a religious motivational speaker and relationship advisor. The talk was centred around positive male representation in the community.

### 2.4. Participants

The participants recruited for both studies were males between the ages of 18 and 35 who were isiZulu speaking. They had to reside in the study areas and had to be available for a follow-up at 3 to 4 months post-intervention. The research participants were recruited from community sites such as schools, churches, and community organizations. In both areas the recruitment drives were conducted to include a well-publicized initiative of talks about the study aims at community meetings, local churches, and sports tournaments organized specifically for this purpose. In the peri-urban area, the research team was also hosted at a local community radio station to answer any questions the community had about the study. These recruitment drives continued for nearly 12 months before commencing the studies.

Participation in the study was on a voluntary basis to all participants who met the inclusion criteria and were able to come and take part during the times allocated. Researchers provided transport (where necessary) and also made provision for refreshments to the participants.

### 2.5. Training Peer Educators

A total of 6 peer educators (2 for rural and 4 for the peri-urban area) were recruited and trained to administer the curriculum. Additionally, 2 research managers also underwent the same training but were to play a supportive role to the research director (lead author). The strategic decision to recruit the research managers was informed by the consultative processes with the communities leading up to the study. It was agreed during the consultative process that if the research managers were older than the peer educators and had actively participated in community upliftment projects, the peer educators would be more motivated to do the work. The training process was also used as an opportunity to further adapt the intervention. These changes included terminology specific for the areas concerned. All peer educators and managers were recruited in the study areas and were fluent in isiZulu.

### 2.6. Study Instrument

Data for both the pre-test and post-test in both studies were collected through a facilitator-administered questionnaire. This questionnaire was adapted from a previous study among male prison inmates in the KwaZulu-Natal and Mpumalanga provinces [31,32]. This study questionnaire was divided into two sections, the first measured the demographic profile of the population in terms of age, level of education, level of income, and employment status, and the second section examined the psychosocial determinants of risky sexual behaviours and substance use, measuring attitudes, subjective norms, perceived behavioural control, and behavioural intentions based on the theory of planned behaviour [26].

The psychosocial determinants of 5 behaviours were measured: using a condom consistently, getting tested for HIV, reducing total alcohol and drug intake, avoiding sex when intoxicated, and avoiding sex with intoxicated people. The theory of planned behaviour variables for attitude, subjective norm, and intention were measured using a scale of 1 to 5 with answering options of: 1 = strongly disagree, 2 = disagree, 3 = unsure, 4 = agree, and 5 = strongly agree, while perceived behavioural control was measured using a scale of 1 to 5 with options of: 1 = very confident, 2 = confident, 3 = unsure, 4 = not confident, and 5 = not confident at all. Table A1 and Table A2 (see Appendix A) provide an overview of the psychosocial concepts that were measured, including the number of items, sample items, minimum and maximum score, mean, standard deviation, and Cronbach’s Alpha (three or more items) as a measure of the internal consistency of grouped items. Note that subjective norm towards condom use was erroneously omitted from data collection and therefore not included in the analyses.

The questionnaire was developed in English and translated into isiZulu, then it was translated back into English to ensure construct and face validity. The research assistants, together with the project managers, who came from the same background as the research participants, were responsible for the translation process. The translations were all done in the form of a workshop with all the research assistants, project managers, and some of the co-authors in attendance. Consensus was reached for the correct use of language for all of the research tools.

### 2.7. Statistical Analysis

Statistical analysis was done using SPSS Version 24 (Statistical Product and Service Solutions, IBM, New York, NY, USA). For study 1, the peri-urban cohort, the first step of the analysis was a one-way analysis of variance (ANOVA) to test for differences in the means at baseline between the experimental and the control group. Then, in the next step, repeated measures analyses were conducted to determine if there was any change in the psychosocial measures scores over time (time × group interaction effect). For study 2, the rural cohort, pairwise samples t-tests were conducted to determine the difference between pre-test and post-test means of the respective variables. Alpha levels for statistical significance for these analyses in both studies were adjusted using the Bonferroni correction (0.05/19 variables = *p* ≤ 0.002). The Bonferroni correction is used to reduce the chances of obtaining type 1 errors when multiple pairwise tests are performed on a single set of data. It should be noted that there are arguments for [33,34,35,36] and against [37] the use of parametric tests in the analysis of ordinal data, as collected for Likert Scales. Some of the rationale against the use of parametric tests for ordinal data is that the data is often not normally distributed. Although this may be true in certain instances, tests such as t-test and ANOVA depend on the normal distribution of means not data and it has been shown that for sample sizes greater than 10 the means approximate normal distribution, irrespective of the original distribution [34]. Nevertheless, non-parametric tests were also performed, which yielded similar results.

### 2.8. Ethical clearance

The study received full ethical clearance from the South African Medical Association Research Ethics Committee (SAMAREC-Protocol MRC 1-09), which works according to the guidelines of the Helsinki Declaration on ethical aspects in human experimentation, and additionally, the research team also sought and received permission from the local municipal offices and the traditional leadership in the area concerned.

## 3. Results

### 3.1. Socio Demographic Profile

A total number of 575 participants completed the pre-test and 439 participants completed the post-test questionnaires in both studies combined. The majority of the participants were between the ages of 18 and 25, with 82.5% for the peri-urban pre-test, 86.4% for the peri-urban post-test, 97.1% for the rural pre-test, and 96.1% for the rural post-test. The level of education was from primary until tertiary, although only a small number reported having studied post-matric. The large majority of the participants were not employed (see Table 1).

### 3.2. Intervention Effects on Psychosocial Determinants

For study 1, Table 2 shows the mean scores at pre-test and post-test for the control and experimental groups. Table 3 shows the results of the one-way ANOVA’s, comparing the mean scores of the peri-urban experimental and peri-urban control group on the outcome measures at baseline. The results show that for 4 out of 19 variables the control and experimental groups scored significantly different at baseline. 

Table 4 reports on the repeated measures ANOVA that was conducted to compare the effect of the behavioural intervention over time in the peri-urban setting. After correction for multiple testing there were no significant time × group interactions for the variables tested. 

For study 2, Table 5 shows the pairwise comparisons, testing the difference between the pre-test scores and post-test scores on the outcome measures. Only one significant difference was shown between baseline and post-test for attitude towards avoiding sex when one is intoxicated, suggesting that there was not much of an effect over time with exposure to the intervention in between. 

## 4. Discussion

The objectives of this study were to test the effectiveness of a behavioural intervention developed to target risky sexual behaviours of lack of correct or consistent condom use, not getting tested for HIV, alcohol and drug use, sexual intercourse when one is intoxicated, and lastly, sexual intercourse with intoxicated people. The intervention was not designed to target the behaviours directly, but to work through effecting change in the psychosocial determinants of attitude, subjective norm, and perceived behavioural control, which would then lead to stronger intention to use condoms, stronger intention to get tested for HIV, stronger intention to reduce alcohol and drug use, stronger intention to avoid sex when one is intoxicated, and stronger intention to avoid sex with intoxicated people. Strong intention towards the target behaviour is likely to yield the desired positive change in that behaviour [38]. Overall, the outcomes of the current intervention show a very minimal effect on the psychosocial determinants tested. 

The results for study 1 among males in a peri-urban area showed a significant difference in 4/19 variables at baseline when comparing control and intervention mean scores. The two groups were quite similar at baseline. It was expected that selection biases would be eliminated at randomization and that there would not be any significant differences at baseline. At post-test, after the participants had gone through the experiment and control conditions, there was no statistical support for the predicted intervention. Given the very positive feedback received from the participants, especially after the condom use and substance use sessions, we had expected a positive increase in the intention to use condoms or intention to reduce alcohol and drug use. There was no effect in either of these measures. This may have been due to the sessions on condom use and alcohol and drug use not being long or detailed enough or skills building components may not have been adequate. Also, social support for condom use may have been lacking as participants had mentioned that some of the nursing staff at public health facilities were at times very unfriendly, which may have negatively affected intention to use a condom. The participants had mentioned that some of the negative comments by the staff regarding the participants’ sexual activities in relation to accessing condoms had discouraged the participants from going back to the clinics. 

The results for study 2 among males in the rural area who participated in the intervention showed a significant increase in attitude towards avoiding sex when one is intoxicated. We would expect that this finding would have been complimented by a similar increase in the measures associated with the reduction of alcohol and drug use, but this was not the case. Additionally, there was also no effect observed in the measures associated with condom use, with getting tested for HIV, and with avoiding sex with intoxicated people. One of the reasons for this intervention not having the desired positive effects, for getting tested for HIV for example, could be due to the fact that the intervention did not have modules designed specifically for strengthening self-efficacy to get tested or to overcome the stigma surrounding HIV testing and the negative attitudes of health care staff. The participants had mentioned levels of distrust they had towards health care staff, saying that they do not trust staff to keep their HIV status confidential within their respective communities. A possible explanation for the lack of impact of the intervention on intention to reduce risky sexual behaviours and alcohol and drug use may also be that the intervention was still too focused on information provision and not enough on increasing self-efficacy and skills of young people to not perform or to reduce these risky behaviours. Strengthening self-efficacy has been shown to have favourable results on positive performance [39]. 

This study targeted young men aged 18–35 for the intervention. The most recent HIV national survey reports that in South Africa, condom use increased for the 25–49 year olds but remained the same for the 15–24 years [1] and the majority of participants in the current study were between 18 and 25 years. To be more effective, follow-up research on this intervention may have to consider including people at a younger age, when risky sexual behaviours and addictions to alcohol and other substances are initiated and established. There is evidence that already suggests that young people are engaging more in alcohol abuse and risky sexual behaviours [40]. The current intervention could then be more effective if it was further adapted and included into programmes for secondary school learners. Recent research shows that using TPB can be used in planning HIV prevention programmes for young people in South Africa [41]. Additionally, using a systematic planning framework with well-designed protocols, like Intervention Mapping, can assist the success of the intervention [42]. 

The current intervention is characterized by sessions that included communications training, skills training, confidence building, role playing, critical dialogue, critical reflection, and peer support. It draws from the thinking that education through dialogue and participation empowers the recipients to question the current state of reality, making them more amenable to an alternative viewpoint [43]. This was particularly evident during the curriculum sessions of the present study, with participants consistently expressing that prior to the introduction of this study they had no outlet for them to discuss these topics with their peers. This excitement may have also contributed to the contamination of the intervention being widely discussed in the community, thereby allowing participants from the control group to know about the topics discussed in the intervention group. For comparative purposes, there are not many studies that utilized this approach in developing health interventions, or specifically, those studies targeting attitudes, social norms, and perceived control with the aim to influence intention towards targeted behaviours. The prison study from which the current study was adapted sought to reduce risky sexual behaviours through peer-facilitated discussions and skills training among soon-to-be-released prison inmates. The prison study, similar to the current study, targeted the determinants of intention that would in turn influence behaviour. It was found that inmates from the experimental group had higher knowledge of STI’s, stronger intention to reduce risky sexual behaviours, and generally more positive self-efficacy and sexual communication [31]. The prison study had two post-test assessments and the longer-term assessment confirmed that indeed there had been a positive behaviour change. Introducing two post-test assessments could perhaps also be explored in future development of the current intervention. 

Another limitation of this study is that the rural cohort did not have a control arm, therefore making it difficult to make direct comparisons with the peri-urban cohort. These comparisons might have shed more light on why only one variable was positively affected. Additionally, if the study had observed or measured the risky behaviours directly, it might have also helped to observe if the risky behaviours had already started to change and whether possible changes were affected by the changes in the proximal cognitive determinants. With regards to contamination, there may have been several possible avenues where this happened. First, the participants completed the questionnaires in a hall-sized room with peer educators present to assist where needed. There may also have been contamination with some participants speaking or checking one another’s responses, even though the research team was cautious to discourage that during the completion of the questionnaires. Secondly, the peer educators who administered the intervention curriculum and those who administered the control curriculum were trained together, it is possible that during the debriefing discussion after administering the control curriculum, the peer educators in the control conditions may have inadvertently introduced intervention-related opinions to the discussions. The reason the research team trained them together was so that all of them would be equipped to replace others should they report sick or be unable to attend their own assigned sessions. Lastly, the participants in the control group and those in the experiment group could have interacted socially in the community over the period of the intervention curriculum. Another important limitation is that our data was self-reported. Self-reported results are prone to social desirability biases. In future, more stringent randomization processes should be used to eliminate selection biases and to strengthen the capacity to make causal inferences on the results.

There was also a considerable number of participants who were lost to follow-up, particularly in study 1. The main reason for a substantial loss was that a number of participants who had been in their final year of high school had relocated in the following year for further studies away from the study area and therefore could not be followed up. Additionally, communication with the participants for reminders about sessions was primarily through their mobile phones and reminders were communicated at least 3 times before the session dates. Despite these efforts, there were still challenges whereby some participants were not reachable at times. Furthermore, since most of the participants were unemployed, the transport that the team had provided was the sole means for them to get to the venue where the sessions were held. Some of the participants reported a loss of interest in the intervention and thus were not willing to participate further. Successful retention of participants needs careful attention in the design of future intervention studies.

## 5. Conclusions

Overall, the intervention had some success, particularly at increasing young men’s attitudes towards avoiding sex when intoxicated. However, no effects were achieved in terms of increasing intentions associated with the other behaviours of condom use, HIV testing, reducing alcohol and drug use, and avoiding sex with intoxicated people. Further development and testing of this programme is recommended before it can be considered for broader scale implementation.

## Figures and Tables

**Table 1 ijerph-16-01938-t001:** Socio-demographic profile of the participants. Study 1 Experiment (*n* = 236), Control (*n* = 192), and Study 2 (all Experiment) (*n* = 147).

Characteristic	Frequency					
	Study 1 (Peri-Urban)				Study 2 (Rural)	
	Pre-Test: Control	Post-Test: Control	Pre-Test: Experiment	Post-Test: Experiment	Pre-Test	Post-Test
**Age** (18–20) (21–25) (26–30) (31–35)	174 (90.6%) 13 (6.8%) 4 (2.1%) 1 (0.5%)	105 (82%) 15 (11.7%) 5 (3.9%) 3 (2.3%)	120 (51.3%) 73 (31.2%) 30 (12.8%) 11(4.7%)	97 (55.7%) 44 (25.3%) 23 (13.2%) 10 (5.7%)	110 (78%) 27 (19.1%) 4 (2.8%) 0	90 (70.9%) 32 (25.2%) 4 (3.1%) 1 (0.8%)
**Levels of education** Primary school High school Matric Tertiary	4 (2.4%) 128 (77.1%) 25 (15.1%) 9 (5.4%)	1 (0.8%) 70 (56.9%) 37 (30.1%) 15 (12.2%)	2 (0.9%) 105 (45.4%) 89 (38.5%) 35 (15.2%)	0 50 (29.1%) 97 (56.4%) 22 (12.8%)	1 (1.4%) 102 (73.7%) 29 (20.7%) 6 (4.2%)	1 (0.8%) 72 (57.6%) 43 (34.4%) 9 (7.2%)
**Employment status** Employed Not employed	5 (2.6%) 186 (97.4%)	5 (3.9%) 122(96.1%)	12 (5.2%) 220 (94.8%)	13 (7.3%) 165 (92.7%)	5 (3.6%) 135(96.4%)	1 (0.8%) 126 (99.2)
**HIV testing** Previously tested for HIV Previously not tested for HIV	23 (12%) 168 (87.5%)	33 (26.2%) 93 (73.8%)	54 (22.9%) 161 (68.2%)	69 (39.7%) 105 (60.3%)	38 (25.9%) 96 (65.3%)	72 (56.3%) 56 (43.8%)

**Table 2 ijerph-16-01938-t002:** Mean scores (M) and standard deviations (SD) for the two groups in Study 1, experiment (*n* = 236) and control (*n* = 192) at pre-and post-test.

Variable	Study 1—Control				Study 1—Experiment			
	Pre-Test		Post-Test		Pre-Test		Post-Test	
	M	SD	M	SD	M	SD	M	SD
Attitude towards condom use	4.54	0.61	4.35	1.06	4.47	.86	4.33	1.04
Perceived behavioural control towards condom use	4.31	0.78	4.28	1.02	4.44	0.81	4.33	0.97
Intention towards condom use	4.11	1.09	4.03	0.94	4.18	1.09	4.05	0.89
Attitude toward getting tested for HIV	4.56	0.70	4.31	1.09	4.53	0.75	4.35	0.94
Subjective norm toward getting tested for HIV	4.08	1.03	3.93	0.82	4.02	1.09	3.88	0.87
Perceived behavioural control toward getting tested for HIV	4.45	0.76	4.15	1.08	4.33	0.78	4.28	0.99
Intention toward getting tested for HIV	4.18	0.99	3.93	0.92	3.93	1.15	3.92	0.91
Attitude towards reducing alcohol and drug use	4.39	0.92	4.19	1.12	4.39	0.92	4.29	1.00
Subjective norm towards reducing alcohol and drug use	3.87	0.99	3.89	0.89	3.81	1.01	3.70	0.89
Perceived behavioural control towards reducing alcohol and drug use	4.30	0.80	4.04	1.12	4.26	0.83	4.13	1.08
Intention towards reducing alcohol and drug use	4.32	0.86	3.98	0.89	4.00	1.10	3.96	0.93
Attitude towards avoiding sex when you are intoxicated	4.36	1.00	4.26	1.07	3.95	1.26	4.21	1.13
Subjective norm towards avoiding sex when you are intoxicated	4.02	0.93	3.85	0.99	3.76	1.06	3.73	0.91
Perceived behavioural control towards avoiding sex when you are intoxicated	4.02	0.74	4.00	1.02	3.90	0.85	4.04	0.96
Intention towards avoiding sex when you are intoxicated	4.24	0.90	3.94	0.94	4.00	1.10	3.95	0.99
Attitude towards avoiding sex with intoxicated people	4.03	1.36	4.08	1.25	3.64	1.49	4.19	1.16
Subjective norm towards avoiding sex with intoxicated people	3.90	1.05	3.75	0.99	3.51	1.23	3.69	0.91
Perceived behavioural control towards avoiding sex with intoxicated people	4.22	0.82	4.27	1.06	4.03	1.01	4.17	1.18
Intention towards avoiding sex with intoxicated people	4.20	1.03	3.97	1.00	3.88	1.22	4.03	0.89

**Table 3 ijerph-16-01938-t003:** Bivariate results of Study 1 experiment and control means at baseline.

Variables	Source	df	Sum of Squares	Mean Square	F	*p*
Attitude towards condom use	Between groups	1	0.752	0.752	1.290	0.257
	Within groups	424	247.125	0.583		
	Total	425	247.877			
Perceived behavioural control towards condom use	Between groups	1	0.252	0.252	0.406	0.525
	Within groups	422	261.999	0.621		
	Total	423	262.251			
Intention towards condom use	Between groups	1	0.390	0.390	0.325	0.569
	Within groups	426	511.865	1.202		
	Total	427	512.255			
Attitude toward getting tested for HIV	Between groups	1	1.067	1.067	2.063	0.152
	Within groups	422	218.200	0.517		
	Total	423	219.266			
Subjective norm toward getting tested for HIV	Between groups	1	0.765	0.765	0.662	0.416
	Within groups	426	492.831	1.157		
	Total	427	493.596			
Perceived behavioural control toward getting tested for HIV	Between groups	1	3.373	3.373	5.666	0.018
	Within groups	423	251.816	0.595		
	Total	424	255.189			
Intention toward getting tested for HIV	Between groups	1	9.202	9.202	7.779	0.006
	Within groups	425	502.757	1.183		
	Total	426	511.959			
Attitude towards reducing alcohol and drug use	Between groups	1	0.071	0.071	0.085	0.770
	Within groups	422	351.797	0.834		
	Total	423	351.868			
Subjective norm towards reducing alcohol and drug use	Between groups	1	1.643	1.643	1.566	0.211
	Within groups	425	445.762	1.049		
	Total	426	447.404			
Perceived behavioural control towards reducing alcohol and drug use	Between groups	1	.887	0.887	1.334	0.249
	Within groups	421	279.837	0.665		
	Total	422	280.724			
Intention towards reducing alcohol and drug use	Between groups	1	9.886	9.886	9.936	0.002 **
	Within groups	426	423.847	0.995		
	Total	427	433.733			
Attitude towards avoiding sex when you are intoxicated	Between groups	1	15.657	15.657	12.166	0.001 **
	Within groups	423	544.367	1.287		
	Total	424	560.024			
Subjective norm towards avoiding sex when you are intoxicated	Between groups	1	4.602	4.602	4.426	0.036
	Within groups	426	442.975	1.040		
	Total	427	447.577			
Perceived behavioural control towards avoiding sex when you are intoxicated	Between groups	1	2.655	2.655	4.007	0.046
	Within groups	425	281.637	0.663		
	Total	426	284.292			
Intention towards avoiding sex when you are intoxicated	Between groups	1	4.340	4.340	3.977	0.047
	Within groups	425	463.831	1.091		
	Total	426	468.171			
Attitude towards avoiding sex with intoxicated people	Between groups	1	16.881	16.881	8.996	0.003
	Within groups	423	793.734	1.876		
	Total	424	810.615			
Subjective norm towards avoiding sex with intoxicated people	Between groups	1	14.134	14.134	10.846	0.001 **
	Within groups	425	553.832	1.303		
	Total	426	567.966			
Perceived behavioural control towards avoiding sex with intoxicated people	Between groups	1	3.834	3.834	4.421	0.036
	Within groups	425	368.610	0.867		
	Total	426	372.445			
Intention towards avoiding sex with intoxicated people	Between groups	1	12.057	12.057	9.554	0.002 **
	Within groups	426	537.566	1.262		
	Total	427	549.623			

** Significant *p* ≤ 0.002 after Bonferroni correction.

**Table 4 ijerph-16-01938-t004:** Repeated Measures Within-Subject Effect, Greenhouse–Geisser (time*group), Study 1 experiment vs control.

Variable	df Hypothesis	df Error	Mean Square	F	*p*	Partial Eta Squared
Attitude towards condom use	1	299	0.074	0.087	0.768	0.000
Perceived behavioural control towards condom use	1	298	0.269	0.354	0.552	0.001
Intention towards condom use	1	299	0.110	0.125	0.724	0.001
Attitude towards getting tested for HIV	1	295	0.166	0.208	0.649	0.001
Subjective norm towards getting tested for HIV	1	298	0.010	0.013	0.909	0.000
Perceived behavioural control towards getting tested for HIV	1	296	2.273	2.752	0.098	0.009
Intention towards getting tested for HIV	1	298	2.006	2.230	0.136	0.007
Attitude towards reducing alcohol and drug use	1	295	0.402	0.406	0.525	0.001
Subjective norm towards reducing alcohol and drug use	1	297	0.527	0.621	0.431	0.002
Perceived behavioural control towards reducing alcohol and drug use	1	295	0.604	0.675	0.412	0.002
Intention towards reducing alcohol and drug use	1	299	2.963	3.431	0.065	0.011
Attitude towards avoiding sex when you are intoxicated	1	298	4.505	3.402	0.066	0.011
Subjective norm towards avoiding sex when you are intoxicated	1	297	0.714	0.783	0.377	0.003
Perceived behavioural control towards avoiding sex when you are intoxicated	1	296	1.025	1.212	0.272	0.004
Intention towards avoiding sex when you are intoxicated	1	298	2.521	2.658	0.104	0.009
Attitude towards avoiding sex with intoxicated people	1	292	9.189	5.031	0.026	0.017
Subjective norm towards avoiding sex with intoxicated people	1	295	4.017	3.699	0.055	0.012
Perceived behavioural control towards avoiding sex with intoxicated people	1	290	0.437	0.847	0.358	0.002
Intention towards avoiding sex with intoxicated people	1	298	5.051	5.432	0.020	0.018

**Table 5 ijerph-16-01938-t005:** Paired samples T-tests showing pre-test versus post-test scores on the variables for the study 2 cohort.

Variable Pair	Mean (SD) Pre-Test	Mean (SD) Post-Test	df	t	*p*
Attitude towards condom use	4.238 (0.885)	4.266 (0.960)	108	−0.218	0.828
Perceived behavioural control towards condom use	4.328 (0.924)	4.200 (0.972)	110	1.008	0.316
Intention towards condom use	3.892 (1.200)	4.031 (0.908)	110	−0.954	0.342
Attitude towards getting tested for HIV	4.268 (0.959)	4.302 (0.951)	108	−0.273	0.785
Subjective norm towards getting tested for HIV	3.727 (1.107)	3.816 (0.862)	109	−0.687	0.494
Perceived behavioural control towards getting tested for HIV	4.297 (0.948)	4.238 (0.981)	109	0.448	0.655
Intention towards getting tested for HIV	3.937 (1.146)	3.890 (0.946)	110	0.317	0.752
Attitude towards reducing alcohol and drug use	4.159 (1.028)	4.247 (0.980)	109	−0.622	0.535
Subjective norm towards reducing alcohol and drug use	3.574 (1.096)	3.592 (0.964)	109	−0.127	0.899
Perceived behavioural control towards reducing alcohol and drug use	4.065 (1.018)	3.927 (1.149)	109	0.953	0.343
Intention towards reducing alcohol and drug use	3.765 (1.270)	3.820 (0.907)	108	−0.343	0.732
Attitude towards avoiding sex when intoxicated	3.448 (1.385)	4.043 (1.134)	108	−3.371	0.001 **
Subjective norm towards avoiding sex when intoxicated	3.355 (1.143)	3.563 (0.992)	110	−1.448	0.150
Perceived behavioural control towards avoiding sex when intoxicated	3.601 (1.001)	3.966 (0.947)	110	−2.878	0.005
Intention towards avoiding sex when intoxicated	3.907 (1.206)	3.829 (0.949)	108	0.504	0.615
Attitude towards avoiding sex with intoxicated people	3.453 (1.487)	3.979 (1.252)	108	−2.783	0.006
Subjective norm towards avoiding sex with intoxicated people	3.438 (1.180)	3.529 (1.023)	109	−0.598	0.551
Perceived behavioural control towards avoiding sex with intoxicated people	3.876 (1.040)	3.957 (1.293)	110	−0.870	0.385
Intention towards avoiding sex with intoxicated people	3.810 (1.211)	3.913 (1.053)	110	−0.658	0.512

** Significant *p* ≤ 0.002 after Bonferroni correction.

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
