# Peer review of "Effectiveness of a Health Behavioural Intervention Aimed at Reduction of Risky Sexual Behaviours among Young Men in the KwaZulu-Natal Province, South Africa"

_ijerph, 2019, doi:10.3390/ijerph16111938_

Round 1

Reviewer 1 Report

The work submitted by Manyaapelo et al. evaluated the effect of behavioral intervention conducted in two areas in KwaZulu-Natal province of South Africa using one study in peri-urban and a second study in rural and found that the intervention had minimal success. It is an interesting work to the readers and please find my comment below.

(1)  The design of study 1 is repeated measures design and please discuss more about the rationale of using one way ANOVA in table 3.  

(2)   In study 1, there are 4 variables which differ at the baseline level between control and experiment groups. Please explain why and how to address this issue in the future studies.

Author Response

Dear Reviewer 1

Thank you for your time and consideration in reviewing our manuscript. Your comments in this revised version have strengthened our article immensely.

We hope that these corrections made as per your suggestions are to your satisfaction and look forward to your final evaluation.

Kindly find the document with a detailed response attached. 

Reviewer 2 Report

This is an interesting and relevant study; the design is in accordance with the objectives and the theoretical framework (The theory of planned behaviour) is adequate. However, the statistical analysis is not correct; The variables used in data analysis are qualitative ordinal variables measured in a Likert scale:

·       variables for attitude, subjective norm and intention were measured using a scale of 1 to 5 with answering options of: 1 = strongly disagree, 2 = disagree, 3 = unsure, 4 = agree, and 5 = strongly agree

·        perceived behavioural control was measured using a scale of 1 to 5 with options of: 1 = very confident; 2 = confident, 3 = unsure; 4 = not confident, and 5 = not confident at all).

With this type of variables, it is not correct to compute means and standard deviations; and it is not adequate to use ANOVA to compare the difference between pre-test and post-test statistics.

The authors must use the median and IQR and to characterize each variable and nonparametric tests  to compare differences between pre-test and post-test Median of each variable.

In summary:

Section on statistical analysis must be presented in a corrected way;  the results must be based on medians and nonparametric tests. Discussion and conclusions must me done in accordance with new results

Author Response

Dear Reviewer 2

Thank you for your time and consideration in reviewing our manuscript. We appreciate the comments for the manuscript. The suggestions are dully noted and will be very much be considered for future studies. We hope that the response given below addressed your concerns adequately and look forward to your final evaluation.

Kindly find the document with a detailed response attached. 

Reviewer 3 Report

-          In the abstract, the authors claim that the intervention has minimal advocacy, so it is unclear whey they promote implementation of the intervention on broader scale before testing efficacy of an adapted version

-          Line 43 – a citation is needed for “key risk behaviors”

-          Line 46, #3 citation provided (Heeren et al) is for adolescent SA males, so it can not be used as a generalized finding for “people”, additional citation has to be provided or it has to be qualified

-          The acronyms STI and HIV are never defined in the manuscript

-          Line 54, the same exact estimate is used for two different years for male s and females from 2017. Please check for accuracy and reformulate the sentence

-          Line 63, there is a semicolon separating two numerical notations for citations (8;9)

-          Line 92/93, there is no closed parentheses

-          In Method’s, what does “African population” infer, are the authors describing race? Because aren’t all participants African because they are natives of SA?

-          Research design should be presented first before details of the intervention are outlined

-          Line 134, “sex aversion during intoxication” is listed as a ‘risky’ behavior

-          How was fidelity assess among peer educators?

-          How many participants were in each session?

-          There seems to be significant lost to follow up (LTFU) for study 1 for example from 236 – 182, almost a quarter of the sample, but the authors do not address reasons for this LTFU

-          Shouldn’t the first column of Table 1 be divided into cases and controls?

-          The table titles are missing information on sample size and setting

-          Table 2, no alpha levels are indicated to highlight significant findings

-          Shouldn’t “sig” column title be “p-level”?

-          No discussion of Bonferroni correction in methods discussion however indicated in the table for study 2

Author Response

Dear Reviewer 3

Thank you for your time and consideration in reviewing our manuscript. We appreciate the comments for the original manuscript, which have really strengthened the content of this now revised version.

Kindly find the document with a detailed response attached. 

Round 2

Reviewer 2 Report

I don’t agree with the justification made by the authors to use ANOVA with variables measured on a Likert scale (that is an ordinal scale).

The arguments made are not very robust; namely the paper cited in  statistical analysis section is not published in a good methodological review (Asian Journal of Psychiatry).

It is very easy to examine a chapter of a standard biostatistics book and to check the conditions underlying parametric tests; and these conditions are not verified with the data presented in this paper. See, for example, chapter 12 of Martin Bland text book (an introduction to medical statistics), where it is clear that if we are analyzing ordinal variables, we must use nonparametric tests.

Bland, M. (2015). An introduction to medical statistics. Oxford University Press (UK).

Author Response

Dear Reviewer 2

We would like to thank you for your time and consideration in reviewing the revised version of our manuscript. We however do not agree that the statistical analyses we used are incorrect.

As stated previously we are aware of how the literature is divided in what tests should be used when analysing data collected using Likert scales. There are articles which specifically address these concerns (Mircioiu & Atkinson, 2017; Murray, 2013; Norman, 2010). What the articles address primarily is that parametric tests can be used on Likert scale data without any fear of getting incorrect results.

Some of the concerns often attributed to why parametric tests should not be used for ordinal data include: sample size, issues about non-normal distribution of Likert scales data, and the robustness of the tests. All these concerns are addressed by the articles we have provided. From the response given in this round, it appears the concerns are twofold, firstly it is where the one article we cited was published (the second article we had provided equally argues for parametric test and is from a reputable MDPI journal) and secondly the concern is that the use of our chosen statistical methods is fundamentally incorrect. We have given two additional citations in the manuscript, which also argue that the use of parametric tests for ordinal data is justified.  The citations we have given adequately address all the methodological concerns often raised by colleagues. For example, the article by Norman 2010 provides a short and comprehensive explanation to support the use of parametric tests.

As per suggestion the suggestion, we also ran the non-parametric analysis for tables 3, 4 and 5. The non-parametric results for table 3 are similar with the ANOVA results, but with the Bonferroni adjustment we get 2/19 (not 4/19) variables showing significant differences at baseline.  For table 4, the non-parametric tests show 1 significant variable (parametric test had none). For table 5, the non-parametric and parametric tests show the exact same significant variables. We have therefore added a line in the methods section which reads:

Lines 431 – 433 - It should be noted that as per recommendations of one of the reviewers, non-parametric test were also performed and yielded similar results.

Again, we appreciate the comments immensely and the suggestions will very much be considered for future studies. We hope that the response given addressed the concerns adequately and look forward to your final evaluation.

Mircioiu, C., & Atkinson, J. (2017). A Comparison of Parametric and Non-Parametric Methods Applied to a Likert Scale. Pharmacy, 5(4), 26. https://doi.org/10.3390/pharmacy5020026

Murray, J. (2013). Likert Data: What to Use, Parametric or Non-Parametric? International Journal of Business and Social Science, 4(11), 258–264.

Norman, G. (2010). Likert scales, levels of measurement and the “laws” of statistics. Advances in Health Sciences Education, 15(5), 625–632. https://doi.org/10.1007/s10459-010-9222-y

Reviewer 3 Report

It was a pleasure to read this revision. All of my initial concerns were adequately addressed. I have one minor comment, which is on  line 119 – Clarify Black African in KwaZulu-Natal province

Author Response

Dear Reviewer 3

We would like to thank you for your time and consideration in reviewing our revised manuscript. Kindly find the response to your suggestion below.

Comments to the authors

It was a pleasure to read this revision. All of my initial concerns were adequately addressed. I have one minor comment, which is on  line 119 – Clarify Black African in KwaZulu-Natal province

Response:

This line was added to explain.  Line 170-171

Black African is the population classification designated for the indigenous groups in South Africa.